# FACT: FREQUENCY-AWARE CHANNEL-GUIDED MULTIVARIATE TIME SERIES FORECASTING

## ABSTRACT

Forecasting Multivariate Time Series (MTS) requires capturing complex intra-channel dynamics and evolving inter-channel dependencies. However, existing methods often struggle to disentangle meaningful signals from inter-channel noise and intricate interaction patterns. To address this, we propose a novel framework that operates entirely in the frequency domain, modeling inter-channel relationships at the component level. Our approach first dynamically decomposes each time series into its constituent frequencies. An Adaptive Band Decomposition mechanism then identifies and isolates the most salient frequency components, simultaneously filtering noise and enhancing computational efficiency. This allows our model to capture time-varying inter-channel dependencies with high fidelity. Furthermore, our learning objective effectively balances accuracy against regularization constraints for both computational efficiency and interpretability. Extensive experiments on diverse, real-world datasets demonstrate that our method achieves competitive performance. Code is available at this repository: https://anonymous.4open.science/r/FACT.

## 1 INTRODUCTION

Multivariate time series (MTS) forecasting supports power scheduling, weather prediction and industrial control, where accuracy, robustness and interpretability are equally critical (Zhou et al., 2021; Wu et al., 2021a; Zhou et al., 2022). Existing research largely falls into two paradigms. Channel-Dependent (CD) models explicitly mix variables but easily introduce spurious correlations and face scalability issues in high dimensions (Zhang & Yan, 2023; Liu et al., 2023; Wang et al., 2023); Channel-Independent (CI) models improve robustness by per-channel processing, but sacrifice genuine couplings and physical interpretability (Nie et al., 2023; Han et al., 2024). This tension indicates a need for fine-grained, controllable interaction modelling.

The core challenge in MTS forecasting lies in disentangling meaningful signals from the noise inherent in complex inter-channel interactions. While spectral analysis offers a promising direction, we observe a critical physical nuance: different spectral components carry distinct semantics—amplitude reflects energy intensity, while phase encodes temporal alignment. For instance, daily load patterns (high frequency) and seasonal trends (low frequency) often exhibit different interaction modes (coordination vs. antagonism). A difficulty arises, however, in effectively modeling these "channel-frequency cells" (Fig. 1). Existing spectral methods (Wu et al., 2023; Yi et al., 2023b) typically rely on global reweighting or fixed decomposition, failing to capture dynamic, cell-level dependencies and, crucially, ignoring the explicit role of phase shifts in causal alignment.

To address this difficulty, we propose **FACT** (**F**requency-**A**daptive **C**omplex **T**ransformer), which shifts interaction modeling from raw channels to specific frequency components. Unlike real-valued approaches that struggle with phase alignment, FACT operates in the complex domain to explicitly model both magnitude coherence $\Gamma$ and phase offsets $\Phi$. Our solution comprises three steps: (i) a Dynamic Frequency-Band Decomposition (DynFBD) that adaptively isolates salient frequency cells; (ii) a ChannelPriorMixer that leverages physical priors $(\Gamma, \Phi)$ to guide interaction; and (iii) a complex-valued fusion mechanism that aligns these priors with the representation. This design ensures that interactions are physically grounded and robust to noise.

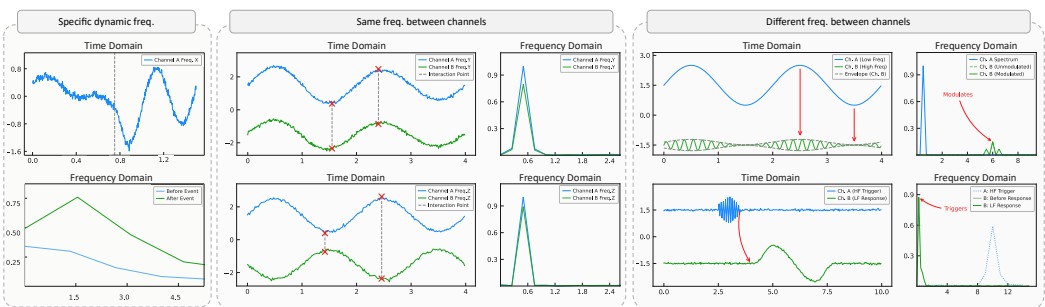

Figure 1: Representative channel–frequency interactions: dynamic drift within a channel (left), same-frequency coordination/antagonism (middle), and cross-frequency modulation/triggering (right, e.g., a sudden cold snap inducing low-frequency heating demand).

- To establish a frequency-level interaction paradigm, we treat the channel–frequency cell as the basic unit and design a sparse token pipeline (DynFBD + selector) to suppress noisy bands while preserving physically meaningful signals.

- We introduce ChannelPriorMixer and adaptive fusion to leverage magnitude/phase-aware priors. By grounding the interaction mechanism directly in physical properties (coherence $\Gamma$ and phase $\Phi$), this design provides intrinsic interpretability, enabling users to trace frequency selection and channel coupling patterns regardless of the chosen backbone.

- Functioning as a model-agnostic plug-in, FACT separates the Frequency-Aware Interaction Module from the representation encoder. This design explicitly prepares frequency-aligned features and can be plugged into diverse backbones (Transformer/MLP/Linear), yielding consistent improvements across datasets compared to raw-channel mixing.

We validate these claims through comprehensive experiments: ablations on each component, regularization sweeps, and interpretability visualizations. Results demonstrate positive correlation between our interpretability metrics and accuracy, and consistent gains across backbones. Details are provided in Section 5.

## 2 RELATED WORK

### 2.1 CHANNEL INTERACTION MODELLING

Early multivariate forecasting adopted RNN/CNN backbones with local dependencies (Hochreiter & Schmidhuber, 1997; Bai et al., 2018), later extended by graph and multi-task formulations that encode handcrafted adjacencies (Wu et al., 2020; 2021b; Cui et al., 2021). Transformers broaden the receptive field (Vaswani et al., 2017; Zhou et al., 2021; Wu et al., 2021a; Zhou et al., 2022), but how to model variable interactions remains contentious. Channel-independent (CI) designs (e.g., PatchTST, iTransformer) favor per-channel tokenization for robustness to noise/drift (Nie et al., 2023; Liu et al., 2023); some even argue high-amplitude frequencies dominate prediction (Dai et al., 2024; Xu et al., 2024). Channel-dependent (CD) methods (Crossformer, CARD, SOFTS, TimePro, DUET) reintroduce interactions via cross-dimension routes, alignment-aware attention, global cores or routing/clustering (Zhang & Yan, 2023; Wang et al., 2023; Han et al., 2024; Ma et al., 2025; Qiu et al., 2025). Recent works like TimeFilter and TQN also explore advanced filtering mechanisms (Hu et al., 2025; Lin et al., 2025), yet they largely rely on spatial-temporal graph filtrations. In contrast, FACT adopts a pure frequency-domain approach to decouple fine-grained interactions. CI may discard genuine couplings; CD often mixes signals coarsely and is sensitive to noise—motivating frequency-aware, fine-grained priors as a middle ground.

### 2.2 TIME–FREQUENCY METHODS AND PHYSICAL PRIORS

Spectral approaches provide efficiency but typically treat amplitude as the sole carrier of information, whereas phase determines temporal alignment/lag and spatial shift. TimeMixer/TimeMixer++

mix frequency bands for long contexts yet collapse phase cues into shared representations (Wang et al.; 2025). FredFormer and TSMixer refine spectra via normalization or MLP mixing, but channel fusion remains entangled and phase alignment implicit (Piao et al., 2024; Ekambaram et al., 2023). FreTS/FITS recalibrate responses (Yi et al., 2023a; Xu et al., 2024), yet they average across channels and cannot reveal which variable drives a specific band or how cross-frequency triggering unfolds. A complementary line emphasizes that spectral components should not be treated uniformly: FreDF shows frequency utility is scenario-dependent and benefits from dynamic fusion (Zhang et al., 2024); periodicity decoupling highlights the role of high-frequency harmonics beyond mere noise (Dai et al., 2024). These observations motivate modelling interactions at the channel–frequency cell with explicit magnitude/phase priors and channel-specific reweighting—precisely what FACT operationalizes. Beyond accuracy, recent work values robustness and interpretability. CI strategies offer stability but little diagnosis (Han et al., 2023); CD designs (SOFTS/CARD) balance the two via global cores or alignment penalties (Han et al., 2024; Wang et al., 2023). FACT inherits spectral efficiency and contributes a physically grounded, fine-grained interaction paradigm that plugs into diverse backbones.

## 3    PRELIMINARIES

**Problem Formulation.** Let $\mathbf{X} = \{\mathbf{x}_1, \ldots, \mathbf{x}_L\} \in \mathbb{R}^{L \times C}$ represent the historical multivariate time series with lookback window $L$ and $C$ channels. The objective is to predict the future sequence $\mathbf{Y} = \{\mathbf{x}_{L+1}, \ldots, \mathbf{x}_{L+T}\} \in \mathbb{R}^{T \times C}$ of length $T$. This forecasting task can be formulated as learning a mapping function $\mathcal{F}_\theta$:

$$\hat{\mathbf{Y}} = \mathcal{F}_\theta(\mathbf{X}), \quad \mathcal{F}_\theta : \mathbb{R}^{L \times C} \to \mathbb{R}^{T \times C}. \tag{1}$$

Our goal is to optimize the parameters $\theta$ such that the predicted $\hat{\mathbf{Y}}$ accurately approximates the ground truth $\mathbf{Y}$, capturing both intra-series temporal dynamics and inter-series channel dependencies.

**Frequency Domain Processing.** To capture global temporal patterns and periodic dependencies, FACT operates in the frequency domain. We apply the real Fast Fourier Transform (rFFT) to the input $\mathbf{X}$ along the time dimension:

$$\mathbf{X}_{\text{fft}} = \mathcal{F}_{\text{rfft}}(\mathbf{X}) \in \mathbb{C}^{F \times C}, \quad F = \lfloor L/2 \rfloor + 1. \tag{2}$$

Unlike methods that process real and imaginary parts separately, we maintain the complex representation in polar form to explicitly preserve physical semantics:

$$\mathbf{X}_{\text{fft}}(f, c) = A(f, c) \cdot e^{i\theta(f,c)}, \tag{3}$$

where $A(f, c) \in \mathbb{R}_{\geq 0}$ denotes the amplitude (representing energy intensity), and $\theta(f, c) \in [-\pi, \pi]$ denotes the phase (representing temporal alignment). This decomposition serves as the foundation for our physics-aware interaction modeling. Full derivations and additional notations are detailed in Appendix F.

## 4    METHODOLOGY

FACT addresses the CI–CD dilemma by modelling interactions at the *channel–frequency* level with explicit magnitude/phase priors. We first outline the pipeline (Fig. 2), then introduce the key modules and the training-time regularizers. Basic notation and operators are given in Section 3.

### 4.1    ARCHITECTURE AND COMPLEXITY OVERVIEW

Figure 2 overviews the pipeline: (i) RevIN normalization and rFFT transformation; (ii) Adaptive Band Decomposition using Gaussian filters to generate frequency bands; (iii) Complex Linear Projection to create multi-scale tokens and extract mask/weight information; (iv) Feature Alignment through cross-attention and gated networks; (v) Complex encoder with coherence ($L_{coh}$) and phase ($L_{phase}$) regularization losses. Note that while Figure 2 depicts a Complex Transformer Encoder, the core Frequency-Aware Interaction Module (steps ii-iv) is backbone-agnostic and can be coupled with MLP or Linear encoders. A concise summary of the per-module complexity is provided in Section 5.3 (Table 3).

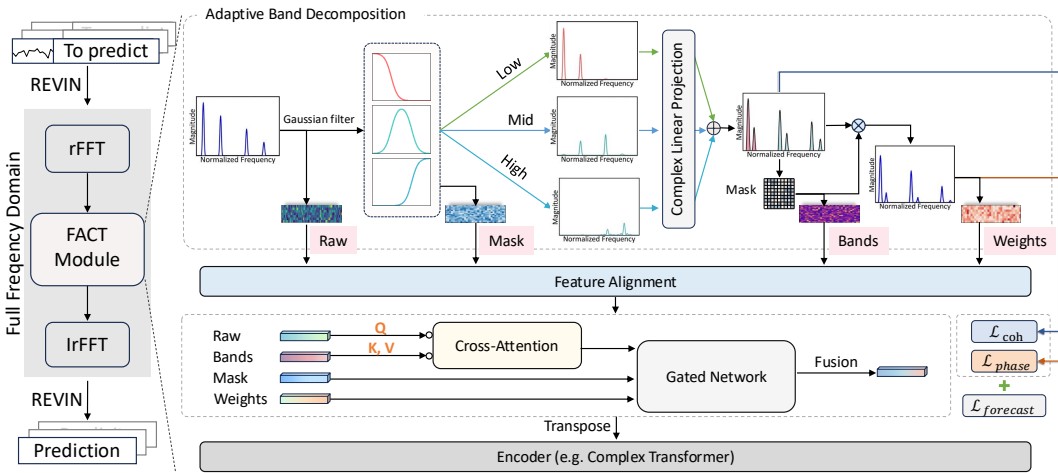

Figure 2: Overall FACT pipeline: input sequences undergo RevIN normalization and rFFT transformation to frequency domain. Gaussian filters perform adaptive band decomposition generating low/mid/high frequency bands, mask, and weight information. Complex linear projection creates multi-scale tokens, followed by Feature Alignment using cross-attention with gated networks. The encoder processes aligned features with coherence and phase regularization losses, finally recovering time-domain predictions through inverse operations.

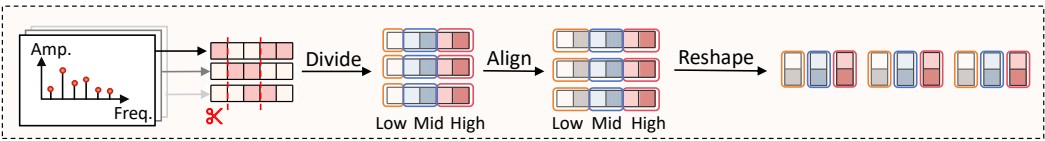

Figure 3: Fixed frequency band division illustration: the frequency axis is divided into low/medium/high three segments according to preset thresholds, each segment is compressed through independent complex linear branches and then concatenated into unified token representation.

## 4.2 ADAPTIVE BAND DECOMPOSITION AND FREQUENCY SELECTION

**Rationale: From Static to Dynamic.** Multi-scale frequencies naturally correspond to seasonalities and lags. A naive approach involves dividing the spectrum into low/mid/high bands using fixed thresholds (see Fig. 3). While this provides a basic interaction unit, it suffers from two limitations: (1) *Energy Truncation*: fixed boundaries may cut through high-energy peaks in diverse datasets (e.g., solar vs. traffic), leading to information loss; (2) *Rigidity*: fixed boundaries lack a mechanism to dynamically re-weight frequency bands and require tedious manual tuning to adapt to different dataset characteristics. To overcome this, we propose an Adaptive Band Decomposition (Fig. 4) driven by learnable Gaussian filters. This design not only softly separates components to avoid aliasing but also produces continuous masks that bridge the frequency frontend with downstream attention modules.

We apply learnable Gaussian filters to each channel to obtain $B_f$ soft frequency bands. Crucially, this process yields both the decomposed tokens $\mathbf{Z}$ and a set of soft masks $\mathbf{P}_{\text{mask}}$:

$$\mathbf{Z}_i = \text{ComplexLinear}(\mathbf{W}_{\text{gauss},i} \odot \mathbf{X}_{\text{fft}}), \quad i = 1, \dots, B_f. \tag{4}$$

The resulting $\mathbf{P}_{\text{mask}}$ and $\mathbf{P}_{\text{weight}}$ are not merely outputs but serve as continuous gating priors injected into the Feature Alignment module (Section 4.5), creating a closed-loop feedback where the model learns to emphasize key frequency bands end-to-end.

The softplus-constrained $(\mu, \sigma)$ parameters are normalized within each band to obtain $(B, C, \text{bands}, F)$ soft masks, which are point-wise multiplied with the original spectrum and pro-

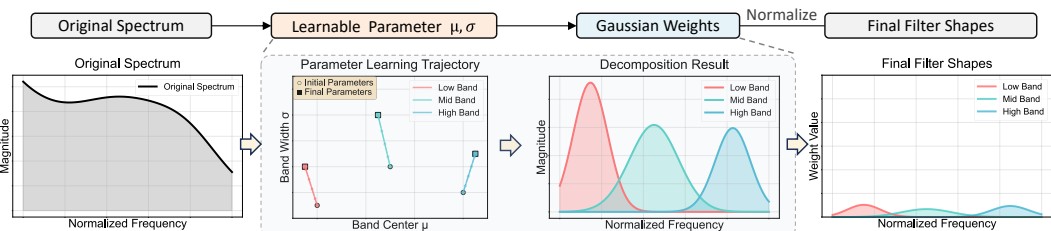

Figure 4: DynFBD's learnable Gaussian filters: raw spectrum, $(\mu, \sigma)$ trajectories, soft-band decomposition, and normalized filter shapes.

jected to $(B, K, 3C)$ via shared complex linear layers. Concurrently, the resulting masks and weights are compressed into low-dimensional summaries $\mathbf{P}^{\text{proj}}_{\text{mask}} \in \mathbb{R}^{B \times F \times d_m}$ and $\mathbf{P}^{\text{proj}}_{\text{weight}} \in \mathbb{R}^{B \times K \times d_w}$, providing interpretable attention bias and gating priors. This soft division not only enables smooth gradients but also forms a closed feedback loop with Feature Alignment, allowing the model to emphasize key frequency bands early in training (see Fig. 4). Empirical results on benchmarks like ETTh1 and ECL show that the Gaussian version reduces sMAPE by approximately $1.3\% \sim 2.1\%$ compared to fixed thresholds.

## 4.3 CHANNEL PRIOR MIXER

**Rationale.** Direct attention on high-dimensional channels is computationally expensive and prone to noise. Moreover, real-valued attention struggles to capture phase-based lead-lag relationships. The Channel Prior Mixer mitigates this by adopting a centralized aggregation-distribution strategy in the complex domain. Specifically, we compute the amplitude coherence $\gamma = \text{Corr}(|\mathbf{X}_{\text{fft}}|)$ and phase difference $\phi = \text{Angle}(\mathbf{X}_{\text{fft}})$ across channels from the input spectrum, serving as the physical ground truth. Based on these priors, we obtain the mixing matrix using learnable scalars $\alpha, \beta$ and temperature $\tau$:

$$\mathbf{M}_{\text{mix}} = \text{softmax}\left(\frac{\alpha\gamma + \beta\phi}{\tau}\right) + \delta\,\mathbf{I}. \tag{5}$$

where $\mathbf{M}_{\text{mix}} \in \mathbb{R}^{C \times C}$. $\mathbf{I}$ is the identity matrix and $\delta$ is a learnable bias to preserve self-channel information. The mixed spectrum is interpolated with strength 0.1, and guided gating compresses amplitudes to $[0, 1]$.

## 4.4 ENCODER PLUGGABILITY

The frequency frontend outputs unified complex tokens, allowing flexibility in the encoder choice based on computational budget: a Complex Transformer (optimal for large channel counts), a Complex MLP (linear cost in $BLd_{\text{model}}d_{\text{ff}}$), or a single-layer Complex Linear (most lightweight). Full comparisons are provided in the Appendix.

## 4.5 FEATURE ALIGNMENT

This module acts as the bridge that injects the physical priors (from Sec 4.3) into the representation stream. Tokens and the raw spectrum are typically misaligned in length and channels. Simple concatenation can cause information leakage and ignore priors. To resolve this, we adopt complex cross-attention where the raw spectrum queries the tokens, while prior-driven gating and bias highlight key bands and suppress noise.

This magnitude–phase pipeline (Fig. 5) allows Feature Alignment to gate strong or weak responses based on amplitude while retaining phase delays, essential for identifying cross-channel lead–lag relations. The module comprises three sub-pathways: (i) query/key projection splitting complex inputs into real/imaginary parts; (ii) value projection preserving phase information; and (iii) a gating generator that learns injection strength and attention bias from mask/weight summaries. The formulation is:

$$\mathbf{Q} = \mathbf{W}_Q[\Re(\mathbf{X}_{\text{fft}}); \Im(\mathbf{X}_{\text{fft}})], \quad \mathbf{K} = \mathbf{W}_K[\Re(\mathbf{Z}); \Im(\mathbf{Z})], \quad \mathbf{V} = \text{ComplexLinear}(\mathbf{Z}). \tag{6}$$

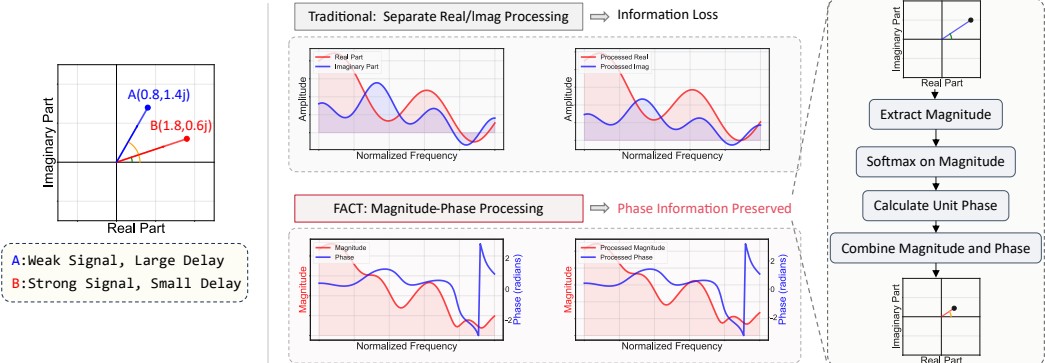

Figure 5: Complex feature handling: traditional real/imaginary split (top) vs. FACT's magnitude–phase processing (bottom). Right: magnitude-softmax and unit-phase reconstruction for complex attention values.

Prior gating and bias are defined as

$$\mathbf{G} = \sigma\big(\mathcal{A}_m(\mathbf{M})\big) \odot \sigma\big(\mathcal{A}_w(\mathbf{W})\big), \quad \mathbf{B} = \mathcal{B}(\mathbf{M}, \mathbf{W}), \tag{7}$$

where $\mathbf{M}, \mathbf{W}$ are projected summaries and $\mathcal{A}_m, \mathcal{A}_w, \mathcal{B}$ are linear mappings. The attention output is

$$\mathbf{H}_{\text{fused}} = \text{Softmax}\Big(\frac{\mathbf{Q}\mathbf{K}^\top}{\sqrt{d}} + \mathbf{B}\Big)\big(\mathbf{V} \odot \mathbf{G}\big). \tag{8}$$

The result is residually interpolated with the original spectrum ($\alpha = 0.7$) and normalized by ComplexLayerNorm. This design maintains $\mathcal{O}(n_{\text{heads}}Kd^2)$ complexity while leveraging prior gating to focus on key frequency bands early in training. Crucially, the cross-attention map ($\mathbf{Q}\mathbf{K}^\top$) in this module serves as a direct visualization window, revealing how the model aggregates multi-scale frequency tokens, thereby providing feature-level interpretability independent of the subsequent encoder backbone.

### 4.6 COMPLEX TRANSFORMER ENCODER

Following frequency-domain alignment, we employ a Complex Transformer Encoder to model long-term dependencies while preserving amplitude-phase information. The encoder consists of two ComplexFullAttentionLayer layers:

$$\mathbf{H}_{\ell+1} = \text{ComplexLayerNorm}\big(\mathbf{H}_\ell + \text{ComplexMultiHeadAttn}(\mathbf{H}_\ell, \mathbf{H}_\ell, \mathbf{H}_\ell)\big), \tag{9}$$

$$\mathbf{H}_{\ell+1} = \text{ComplexLayerNorm}\big(\mathbf{H}_{\ell+1} + \text{ComplexConv1d}(\mathbf{H}_{\ell+1})\big). \tag{10}$$

ComplexMultiHeadAttn reuses weights from Equation equation 6 with prior bias, and ComplexConv1d performs depthwise separable convolution to capture local smoothness. The output is mapped back to $\mathbb{C}^{F \times C}$, then recovered to time-domain predictions through irFFT and inverse normalization.

### 4.7 INTERPRETABILITY REGULARIZATION

To align the model with physical mechanisms during optimization, we impose constraints on cached attention, gating, and priors. This avoids the "train first, interpret later" disconnect. specifically, we cache fusion representations $\hat{\mathbf{H}}$, gating vectors $\mathbf{g}$, mixing matrices $\mathbf{M}_{\text{mix}}$, and frequency-domain phases. Averaging these over the frequency dimension yields amplitude correlations $\hat{\gamma}$ and mean phase differences $\widehat{\Delta\theta}$. These drive the coherence and phase regularizers:

$$L_{\text{coh}} = \|\hat{\gamma} - \gamma\|_2^2, \quad \hat{\gamma} = \text{corr}\left(|\hat{\mathbf{H}}|\right), \tag{11}$$

$$L_{\text{phase}} = 1 - \cos\left(\widehat{\Delta\theta} - \phi\right), \tag{12}$$

Table 1: Multivariate Long-term Forecasting results with prediction lengths $H \in \{96, 192, 336, 720\}$ and fixed lookback window length $L = 96$. The results are taken from SOFTS and iTransformer (Liu et al., 2023).

| Models | FACT (ours) | | SOFTS | | iTransformer | | PatchTST | | TSMixer | | Crossformer | | TiDE | | TimesNet | | DLinear | | SCINet | | FEDformer | |
|---|---|---|---|---|---|---|---|---|---|---|---|---|---|---|---|---|---|---|---|---|---|---|
| Metric | MSE | MAE | MSE | MAE | MSE | MAE | MSE | MAE | MSE | MAE | MSE | MAE | MSE | MAE | MSE | MAE | MSE | MAE | MSE | MAE | MSE | MAE |
| ETTm1 96 | 0.327* | 0.361 | 0.325 | 0.361 | 0.334 | 0.368 | 0.329 | 0.365 | 0.323 | 0.363* | 0.404 | 0.426 | 0.364 | 0.387 | 0.338 | 0.375 | 0.345 | 0.372 | 0.418 | 0.438 | 0.379 | 0.419 |
| ETTm1 192 | 0.376* | 0.392 | 0.375 | 0.389 | 0.377 | 0.391 | 0.380 | 0.394 | 0.376* | 0.392 | 0.450 | 0.451 | 0.398 | 0.404 | 0.374 | 0.387 | 0.380 | 0.389 | 0.439 | 0.450 | 0.426 | 0.441 |
| ETTm1 336 | 0.422 | 0.418 | 0.405 | 0.412* | 0.426 | 0.420 | 0.400 | 0.410 | 0.407* | 0.413 | 0.532 | 0.515 | 0.428 | 0.425 | 0.410 | 0.411 | 0.413 | 0.413 | 0.490 | 0.485 | 0.445 | 0.459 |
| ETTm1 720 | 0.502 | 0.463 | 0.466 | 0.447 | 0.491 | 0.459 | 0.475* | 0.453* | 0.485 | 0.459 | 0.666 | 0.589 | 0.487 | 0.461 | 0.478 | 0.450 | 0.474 | 0.453* | 0.595 | 0.550 | 0.543 | 0.490 |
| ETTm1 Avg | 0.407 | 0.409 | 0.393 | 0.403 | 0.407 | 0.410 | 0.396 | 0.406 | 0.398* | 0.407 | 0.513 | 0.496 | 0.419 | 0.419 | 0.400 | 0.406 | 0.403 | 0.407 | 0.485 | 0.481 | 0.448 | 0.452 |
| ETTm2 96 | 0.193 | 0.275 | 0.180 | 0.261 | 0.180 | 0.264 | 0.184 | 0.264 | 0.182* | 0.266 | 0.287 | 0.366 | 0.207 | 0.305 | 0.187 | 0.267 | 0.193 | 0.292 | 0.286 | 0.377 | 0.203 | 0.287 |
| ETTm2 192 | 0.271 | 0.329 | 0.246 | 0.306 | 0.250 | 0.309* | 0.246 | 0.306 | 0.249* | 0.309* | 0.414 | 0.492 | 0.290 | 0.364 | 0.249* | 0.309* | 0.284 | 0.362 | 0.399 | 0.445 | 0.269 | 0.328 |
| ETTm2 336 | 0.312 | 0.349 | 0.319 | 0.352 | 0.311* | 0.348* | 0.309 | 0.346 | 0.309 | 0.347 | 0.597 | 0.542 | 0.377 | 0.422 | 0.321 | 0.351 | 0.369 | 0.427 | 0.637 | 0.591 | 0.325 | 0.366 |
| ETTm2 720 | 0.417 | 0.408 | 0.405 | 0.401 | 0.412 | 0.407 | 0.409* | 0.402 | 0.416 | 0.408 | 1.730 | 1.042 | 0.558 | 0.524 | 0.408 | 0.403* | 0.554 | 0.522 | 0.960 | 0.735 | 0.421 | 0.415 |
| ETTm2 Avg | 0.298 | 0.340 | 0.287 | 0.330 | 0.288* | 0.332* | 0.287 | 0.330 | 0.289 | 0.333 | 0.757 | 0.610 | 0.358 | 0.404 | 0.291 | 0.333 | 0.350 | 0.401 | 0.571 | 0.537 | 0.305 | 0.349 |
| ETTh1 96 | 0.384* | 0.404 | 0.381 | 0.399 | 0.386 | 0.405 | 0.394 | 0.406 | 0.401 | 0.412 | 0.423 | 0.448 | 0.479 | 0.464 | 0.384* | 0.402* | 0.386 | 0.400 | 0.654 | 0.599 | 0.376 | 0.419 |
| ETTh1 192 | 0.436* | 0.436 | 0.435 | 0.431 | 0.441 | 0.436 | 0.440 | 0.435 | 0.452 | 0.442 | 0.471 | 0.474 | 0.525 | 0.492 | 0.436* | 0.429 | 0.437 | 0.432* | 0.719 | 0.631 | 0.420 | 0.448 |
| ETTh1 336 | 0.480 | 0.458 | 0.480 | 0.452 | 0.487 | 0.458 | 0.491 | 0.462 | 0.492 | 0.463 | 0.570 | 0.546 | 0.565 | 0.515 | 0.491 | 0.469 | 0.481 | 0.459 | 0.778 | 0.659 | 0.459 | 0.465 |
| ETTh1 720 | 0.504 | 0.486 | 0.499 | 0.488* | 0.503* | 0.491 | 0.487 | 0.479 | 0.507 | 0.490 | 0.653 | 0.621 | 0.594 | 0.558 | 0.521 | 0.500 | 0.519 | 0.516 | 0.836 | 0.699 | 0.506 | 0.507 |
| ETTh1 Avg | 0.451* | 0.446 | 0.449 | 0.442 | 0.454 | 0.447 | 0.453 | 0.446 | 0.463 | 0.452 | 0.529 | 0.522 | 0.541 | 0.507 | 0.458 | 0.450 | 0.456 | 0.452 | 0.747 | 0.647 | 0.440 | 0.460 |
| ETTh2 96 | 0.307 | 0.356 | 0.297 | 0.347 | 0.297 | 0.349* | 0.288 | 0.340 | 0.319 | 0.361 | 0.745 | 0.584 | 0.400 | 0.440 | 0.340 | 0.374 | 0.333 | 0.387 | 0.707 | 0.621 | 0.358 | 0.397 |
| ETTh2 192 | 0.383 | 0.400* | 0.373 | 0.394 | 0.380* | 0.400* | 0.376 | 0.395 | 0.402 | 0.410 | 0.877 | 0.656 | 0.528 | 0.509 | 0.402 | 0.414 | 0.477 | 0.476 | 0.860 | 0.689 | 0.429 | 0.439 |
| ETTh2 336 | 0.422 | 0.430 | 0.410 | 0.426 | 0.428* | 0.432* | 0.440 | 0.451 | 0.444 | 0.446 | 1.043 | 0.731 | 0.643 | 0.571 | 0.452 | 0.452 | 0.594 | 0.541 | 1 | 0.744 | 0.496 | 0.487 |
| ETTh2 720 | 0.422 | 0.442 | 0.411 | 0.433 | 0.427* | 0.445* | 0.436 | 0.453 | 0.441 | 0.450 | 1.104 | 0.763 | 0.874 | 0.679 | 0.462 | 0.468 | 0.831 | 0.657 | 1.249 | 0.838 | 0.463 | 0.474 |
| ETTh2 Avg | 0.383 | 0.407 | 0.373 | 0.400 | 0.383 | 0.407 | 0.385 | 0.410 | 0.401 | 0.417 | 0.942 | 0.684 | 0.611 | 0.550 | 0.414 | 0.427 | 0.559 | 0.515 | 0.954 | 0.723 | 0.437 | 0.449 |
| ECL 96 | 0.146 | 0.241* | 0.143 | 0.233 | 0.148* | 0.240 | 0.164 | 0.251 | 0.157 | 0.260 | 0.219 | 0.314 | 0.237 | 0.329 | 0.168 | 0.272 | 0.197 | 0.282 | 0.247 | 0.345 | 0.193 | 0.308 |
| ECL 192 | 0.178 | 0.268 | 0.158 | 0.248 | 0.162 | 0.253 | 0.173* | 0.262* | 0.173* | 0.274 | 0.231 | 0.322 | 0.236 | 0.330 | 0.184 | 0.289 | 0.196 | 0.285 | 0.257 | 0.355 | 0.201 | 0.315 |
| ECL 336 | 0.187* | 0.280 | 0.178 | 0.269 | 0.178 | 0.269 | 0.190 | 0.279* | 0.192 | 0.295 | 0.246 | 0.337 | 0.249 | 0.344 | 0.198 | 0.300 | 0.209 | 0.301 | 0.269 | 0.369 | 0.214 | 0.329 |
| ECL 720 | 0.206 | 0.300 | 0.218 | 0.305 | 0.225 | 0.317 | 0.230 | 0.313* | 0.223 | 0.318 | 0.280 | 0.363 | 0.284 | 0.373 | 0.220* | 0.320 | 0.245 | 0.333 | 0.299 | 0.390 | 0.246 | 0.355 |
| ECL Avg | 0.179* | 0.272* | 0.174 | 0.264 | 0.178 | 0.270 | 0.189 | 0.276 | 0.186 | 0.287 | 0.244 | 0.334 | 0.251 | 0.344 | 0.192 | 0.295 | 0.212 | 0.300 | 0.268 | 0.365 | 0.214 | 0.327 |
| Traffic 96 | 0.409* | 0.273 | 0.376 | 0.251 | 0.395 | 0.268 | 0.427 | 0.272* | 0.493 | 0.336 | 0.522 | 0.290 | 0.805 | 0.493 | 0.593 | 0.321 | 0.650 | 0.396 | 0.788 | 0.499 | 0.587 | 0.366 |
| Traffic 192 | 0.427* | 0.279* | 0.398 | 0.261 | 0.417 | 0.276 | 0.454 | 0.289 | 0.497 | 0.351 | 0.530 | 0.293 | 0.756 | 0.474 | 0.617 | 0.336 | 0.598 | 0.370 | 0.789 | 0.505 | 0.604 | 0.373 |
| Traffic 336 | 0.465 | 0.294 | 0.415 | 0.269 | 0.433 | 0.283* | 0.450* | 0.282 | 0.528 | 0.361 | 0.558 | 0.305 | 0.762 | 0.477 | 0.629 | 0.336 | 0.605 | 0.373 | 0.797 | 0.508 | 0.621 | 0.383 |
| Traffic 720 | 0.512 | 0.315 | 0.447 | 0.287 | 0.467 | 0.302* | 0.484* | 0.301 | 0.569 | 0.380 | 0.589 | 0.328 | 0.719 | 0.449 | 0.640 | 0.350 | 0.645 | 0.394 | 0.841 | 0.523 | 0.626 | 0.382 |
| Traffic Avg | 0.453* | 0.290 | 0.409 | 0.267 | 0.428 | 0.282 | 0.454 | 0.286* | 0.522 | 0.357 | 0.550 | 0.304 | 0.760 | 0.473 | 0.620 | 0.336 | 0.625 | 0.383 | 0.804 | 0.509 | 0.610 | 0.376 |
| Weather 96 | 0.167 | 0.213* | 0.166 | 0.208 | 0.174 | 0.214 | 0.176 | 0.217 | 0.166 | 0.210 | 0.158 | 0.230 | 0.202 | 0.261 | 0.172 | 0.220 | 0.196 | 0.255 | 0.221 | 0.306 | 0.217 | 0.296 |
| Weather 192 | 0.214 | 0.255* | 0.217 | 0.253 | 0.221 | 0.254 | 0.221 | 0.256 | 0.215* | 0.256 | 0.206 | 0.277 | 0.242 | 0.298 | 0.219 | 0.261 | 0.237 | 0.296 | 0.261 | 0.340 | 0.276 | 0.336 |
| Weather 336 | 0.273 | 0.299* | 0.282 | 0.300 | 0.278 | 0.296 | 0.275* | 0.296 | 0.287 | 0.300 | 0.272 | 0.335 | 0.287 | 0.335 | 0.280 | 0.306 | 0.283 | 0.335 | 0.309 | 0.378 | 0.339 | 0.380 |
| Weather 720 | 0.350 | 0.349 | 0.356 | 0.351 | 0.358 | 0.347 | 0.352 | 0.346 | 0.355 | 0.348* | 0.398 | 0.418 | 0.351* | 0.386 | 0.365 | 0.359 | 0.345 | 0.345 | 0.377 | 0.427 | 0.403 | 0.428 |
| Weather Avg | 0.251 | 0.279* | 0.255 | 0.278 | 0.258 | 0.278 | 0.256* | 0.279* | 0.256* | 0.279* | 0.259 | 0.315 | 0.271 | 0.320 | 0.259 | 0.287 | 0.265 | 0.317 | 0.292 | 0.363 | 0.309 | 0.360 |
| Solar 96 | 0.192 | 0.236 | 0.200 | 0.230 | 0.203* | 0.237* | 0.205 | 0.246 | 0.221 | 0.275 | 0.310 | 0.331 | 0.312 | 0.399 | 0.250 | 0.292 | 0.290 | 0.378 | 0.237 | 0.344 | 0.242 | 0.342 |
| Solar 192 | 0.233 | 0.269 | 0.229 | 0.253 | 0.233 | 0.261 | 0.237 | 0.267* | 0.268 | 0.306 | 0.734 | 0.725 | 0.339 | 0.416 | 0.296 | 0.318 | 0.320 | 0.398 | 0.280 | 0.380 | 0.285 | 0.380 |
| Solar 336 | 0.240 | 0.275* | 0.243 | 0.269 | 0.248* | 0.273 | 0.250 | 0.276 | 0.272 | 0.294 | 0.750 | 0.735 | 0.368 | 0.430 | 0.319 | 0.330 | 0.353 | 0.415 | 0.304 | 0.389 | 0.282 | 0.376 |
| Solar 720 | 0.251* | 0.280 | 0.245 | 0.272 | 0.249 | 0.275 | 0.252 | 0.275 | 0.281 | 0.313 | 0.769 | 0.765 | 0.370 | 0.425 | 0.338 | 0.337 | 0.356 | 0.413 | 0.308 | 0.385 | 0.357 | 0.427 |
| Solar Avg | 0.229 | 0.265* | 0.229 | 0.256 | 0.233* | 0.262 | 0.236 | 0.266 | 0.260 | 0.297 | 0.641 | 0.639 | 0.347 | 0.417 | 0.301 | 0.319 | 0.330 | 0.401 | 0.282 | 0.375 | 0.291 | 0.381 |
| Count (1st) | 3 | 2 | 16 | 23 | 2 | 2 | 5 | 7 | 1 | 0 | 3 | 0 | 0 | 0 | 1 | 2 | 1 | 0 | 0 | 0 | 3 | 0 |
| Count (2nd) | 8 | 5 | 12 | 4 | 8 | 11 | 1 | 6 | 2 | 2 | 0 | 0 | 0 | 0 | 1 | 2 | 1 | 2 | 0 | 0 | 0 | 0 |
| Count (3rd) | 8 | 7 | 0 | 2 | 8 | 9 | 6 | 6 | 6 | 3 | 0 | 0 | 1 | 0 | 4 | 3 | 0 | 2 | 0 | 0 | 0 | 0 |

where $\gamma$ and $\phi$ are derived from amplitude/phase priors. The total loss is $\mathcal{L} = \mathcal{L}_{\text{forecast}} + \lambda_{\text{coh}} L_{\text{coh}} + \lambda_{\text{phase}} L_{\text{phase}}$. By composing Adaptive Band Decomposition, channel priors, and regularized complex encoding, FACT achieves both high accuracy and physical interpretability.

## 5 EXPERIMENTS

### 5.1 DATASETS

We follow the public SOFTS benchmarks (Han et al., 2024): ETT (4 subsets), Traffic, Electricity, Weather, Solar-Energy, and PEMS (4 subsets). These cover electricity, transportation and energy scenarios with heterogeneous channels and sampling rates. Full statistics (channels, horizons, splits, sampling) are provided in Appendix E (Table 8).

### 5.2 TRAINING AND IMPLEMENTATION SETTINGS

Key hyperparameters (optimizer, depth, hidden size, subset protocol) are summarized in Appendix (Section C).

### 5.3 MAIN RESULTS AND ABLATION

We evaluate our method against a comprehensive set of baselines, including linear/MLP models (DLinear, TSMixer, TiDE), Transformers (FEDformer, Stationary, PatchTST, Crossformer, iTransformer), and CNN-based approaches (SCINet, TimesNet). Following standard long-sequence protocols (Zhou et al., 2021; Liu et al., 2022), we fix the lookback window to $L = 96$ and report MSE/MAE across standard horizons. Full implementation details are provided in Appendix C.

Table 2: Multivariate Short-term Forecasting results on PEMS datasets with prediction lengths $H \in \{12, 24, 48, 96\}$ and fixed lookback window length $L = 96$.

| Models | | FACT (ours) | | SOFTS | | iTransformer | | PatchTST | | TSMixer | | Crossformer | | TiDE | | TimesNet | | DLinear | | SCINet | | FEDformer | |
|---|---|---|---|---|---|---|---|---|---|---|---|---|---|---|---|---|---|---|---|---|---|---|---|
| Metric | | MSE | MAE | MSE | MAE | MSE | MAE | MSE | MAE | MSE | MAE | MSE | MAE | MSE | MAE | MSE | MAE | MSE | MAE | MSE | MAE | MSE | MAE |
| PEMS03 | 12 | **0.063** | 0.166 | 0.064 | 0.165 | 0.071 | 0.174 | 0.073 | 0.186 | 0.075 | 0.186 | 0.090 | 0.203 | 0.178 | 0.305 | 0.085 | 0.192 | 0.122 | 0.243 | 0.066* | 0.172* | 0.126 | 0.251 |
| | 24 | 0.084 | 0.191 | **0.083** | **0.188** | 0.093 | 0.201 | 0.105 | 0.212 | 0.095 | 0.210 | 0.121 | 0.240 | 0.257 | 0.371 | 0.118 | 0.223 | 0.201 | 0.317 | 0.085* | 0.198* | 0.149 | 0.275 |
| | 48 | 0.127 | 0.234 | 0.114 | **0.223** | 0.125* | 0.236* | 0.159 | 0.264 | 0.121 | 0.240 | 0.202 | 0.317 | 0.379 | 0.463 | 0.155 | 0.260 | 0.333 | 0.425 | 0.127 | 0.238 | 0.227 | 0.348 |
| | 96 | 0.191 | 0.296 | **0.156** | **0.264** | 0.164 | 0.275 | 0.210 | 0.305 | 0.184 | 0.295 | 0.262 | 0.367 | 0.490 | 0.539 | 0.228 | 0.317 | 0.457 | 0.515 | 0.178* | 0.287* | 0.348 | 0.434 |
| | Avg | 0.116 | 0.222* | **0.104** | **0.210** | 0.113 | 0.221 | 0.137 | 0.240 | 0.119 | 0.233 | 0.169 | 0.281 | 0.326 | 0.419 | 0.147 | 0.248 | 0.278 | 0.375 | 0.114* | 0.224 | 0.213 | 0.327 |
| PEMS04 | 12 | 0.075* | 0.179* | 0.074 | **0.176** | 0.078 | 0.183 | 0.085 | 0.189 | 0.079 | 0.188 | 0.098 | 0.218 | 0.219 | 0.340 | 0.087 | 0.195 | 0.148 | 0.272 | **0.073** | 0.177 | 0.138 | 0.262 |
| | 24 | 0.091 | 0.200* | 0.088 | 0.194 | 0.095 | 0.205 | 0.115 | 0.222 | 0.089* | 0.201 | 0.131 | 0.256 | 0.292 | 0.398 | 0.103 | 0.215 | 0.224 | 0.340 | **0.084** | **0.193** | 0.177 | 0.293 |
| | 48 | 0.118 | 0.233 | 0.110 | 0.219 | 0.120 | 0.233 | 0.167 | 0.273 | 0.111* | 0.222* | 0.205 | 0.326 | 0.409 | 0.478 | 0.136 | 0.250 | 0.355 | 0.437 | **0.099** | **0.211** | 0.270 | 0.368 |
| | 96 | 0.162 | 0.280 | 0.135* | 0.244 | 0.150 | 0.262 | 0.211 | 0.310 | 0.133 | 0.247* | 0.402 | 0.457 | 0.492 | 0.532 | 0.190 | 0.303 | 0.452 | 0.504 | **0.114** | **0.227** | 0.341 | 0.427 |
| | Avg | 0.111 | 0.223 | 0.102 | 0.208 | 0.111 | 0.221 | 0.145 | 0.249 | 0.103* | 0.215* | 0.209 | 0.314 | 0.353 | 0.437 | 0.129 | 0.241 | 0.295 | 0.388 | **0.092** | **0.202** | 0.231 | 0.337 |
| PEMS07 | 12 | **0.056** | **0.150** | 0.057 | 0.152 | 0.067* | 0.165 | 0.068 | 0.163* | 0.073 | 0.181 | 0.094 | 0.200 | 0.173 | 0.304 | 0.082 | 0.181 | 0.115 | 0.242 | 0.068 | 0.171 | 0.109 | 0.225 |
| | 24 | **0.072** | **0.168** | 0.073 | 0.173 | 0.088* | 0.190* | 0.102 | 0.201 | 0.090 | 0.199 | 0.139 | 0.247 | 0.271 | 0.383 | 0.101 | 0.204 | 0.210 | 0.329 | 0.119 | 0.225 | 0.125 | 0.244 |
| | 48 | 0.098 | 0.196 | **0.096** | **0.195** | 0.110* | 0.215* | 0.170 | 0.261 | 0.124 | 0.231 | 0.141 | 0.369 | 0.446 | 0.495 | 0.134 | 0.238 | 0.398 | 0.458 | 0.149 | 0.237 | 0.165 | 0.288 |
| | 96 | 0.133 | 0.227 | **0.120** | **0.218** | 0.139* | 0.245 | 0.236 | 0.308 | 0.163 | 0.255 | 0.396 | 0.442 | 0.628 | 0.577 | 0.181 | 0.279 | 0.594 | 0.553 | 0.141 | 0.234* | 0.262 | 0.376 |
| | Avg | 0.090 | 0.185 | **0.087** | **0.184** | 0.101* | 0.204* | 0.144 | 0.233 | 0.112 | 0.217 | 0.235 | 0.315 | 0.380 | 0.440 | 0.124 | 0.225 | 0.329 | 0.395 | 0.119 | 0.234 | 0.165 | 0.283 |
| PEMS08 | 12 | 0.074 | 0.173 | 0.074 | **0.171** | 0.079* | 0.182* | 0.098 | 0.205 | 0.083 | 0.189 | 0.165 | 0.214 | 0.227 | 0.343 | 0.112 | 0.212 | 0.154 | 0.276 | **0.068** | 0.184 | 0.173 | 0.273 |
| | 24 | **0.098** | **0.198** | 0.104 | 0.201 | 0.115* | 0.219* | 0.162 | 0.266 | 0.117 | 0.226 | 0.215 | 0.260 | 0.318 | 0.409 | 0.141 | 0.238 | 0.248 | 0.353 | 0.122 | 0.221 | 0.210 | 0.301 |
| | 48 | **0.149** | **0.241** | 0.164 | 0.253* | 0.186* | 0.235 | 0.238 | 0.311 | 0.196 | 0.299 | 0.315 | 0.355 | 0.497 | 0.510 | 0.198 | 0.283 | 0.440 | 0.470 | 0.189 | 0.270 | 0.320 | 0.394 |
| | 96 | 0.265 | 0.307 | **0.211** | **0.253** | 0.221 | 0.267 | 0.303 | 0.318 | 0.266 | 0.331 | 0.377 | 0.397 | 0.721 | 0.592 | 0.320 | 0.351 | 0.674 | 0.565 | 0.236* | 0.300* | 0.442 | 0.465 |
| | Avg | 0.147 | 0.230* | **0.138** | **0.219** | 0.150* | 0.226 | 0.200 | 0.275 | 0.165 | 0.261 | 0.268 | 0.307 | 0.441 | 0.464 | 0.193 | 0.271 | 0.379 | 0.416 | 0.158 | 0.244 | 0.286 | 0.358 |
| Count (1st) | | 6 | 3 | 7 | 9 | 0 | 1 | 0 | 0 | 0 | 0 | 0 | 0 | 0 | 0 | 0 | 0 | 0 | 0 | 4 | 3 | 0 | 0 |
| Count (2nd) | | 3 | 7 | 8 | 6 | 2 | 2 | 0 | 0 | 2 | 0 | 0 | 0 | 0 | 0 | 0 | 0 | 0 | 0 | 1 | 0 | 0 | 0 |
| Count (3rd) | | 1 | 2 | 1 | 1 | 8 | 5 | 0 | 1 | 2 | 2 | 0 | 0 | 0 | 0 | 0 | 0 | 0 | 0 | 4 | 5 | 0 | 0 |

Tables 1 and 2 summarize the performance across 12 datasets. FACT exhibits distinct superiority on periodic datasets (e.g., Solar-Energy, Weather), validating that our complex-valued modeling effectively captures physical phase shifts often overlooked by baselines. Compared to Channel-Independent methods like PatchTST, FACT better recovers cross-channel coupling, leading to lower errors on highly correlated data like ECL. On PEMS, it remains competitive against specialized spatio-temporal models by inferring latent spatial dependencies via channel coherence, demonstrating robust generalization without pre-defined graph structures. While high-channel regimes like Traffic indicate room for further scaling, the results collectively validate FACT's effectiveness.

The results in Tables 1 and 2 demonstrate several key findings: (1) FACT achieves strong performance across diverse datasets, particularly excelling on Solar-Energy and Weather forecasting tasks; (2) The frequency-domain approach proves effective for capturing temporal dependencies while maintaining computational efficiency; (3) FACT's interpretable design does not compromise prediction accuracy, establishing a favorable trade-off between performance and explainability in multivariate time series forecasting.

**Analysis of Domain Sensitivity.** FACT exhibits distinct superiority on Solar and Weather datasets (ranking 1st in almost all metrics). This aligns with the physical nature of these domains: they are dominated by strong periodicity and cross-channel phase shifts (e.g., solar irradiance delays due to geographical longitude). FACT's complex-valued modeling explicitly captures these phase differences ($\phi$) and amplitude correlations ($\gamma$) via the Channel Prior Mixer, offering an inductive bias that real-valued models (like iTransformer) lack. Conversely, on datasets with irregular load spikes (e.g., ETT), the advantage of frequency decomposition is less pronounced, though FACT remains competitive.

**Efficiency and Ablation Analysis.** To further quantify the contribution of each module and the efficiency of our design, we conducted detailed ablation studies on the Solar and Weather datasets. We also explored alternative designs during development: notably, replacing our complex-valued pipeline with a simple 2-channel real-valued concatenation resulted in inferior performance (approx. 5% degradation on Solar), as it failed to explicitly capture the phase-based lead-lag relationships critical for periodic data. As shown in Table 4, removing the Dynamic Frequency Band Decomposition (DynFBD) leads to a performance drop, confirming the importance of frequency disentanglement. Crucially, our Adaptive Fusion mechanism demonstrates superior scalability: on the high-dimensional Electricity dataset (321 channels), it reduces computational overhead by over **82%** (10.23s vs. 58.55s per epoch) compared to the concatenation baseline (FACT-concat), which required a reduced batch size to avoid memory overflow. This validates the efficiency of our "filter-then-fuse" strategy for large-scale applications.

We further analyze the theoretical complexity of each module in Table 3. FACT maintains a favorable efficiency profile; the channel mixer operates on top-$k$ bands with linear dependence on

channels $\mathcal{O}(Ck)$, while the adaptive fusion scales with $\mathcal{O}(Kd^2)$, avoiding quadratic complexity w.r.t sequence length $L$.

Table 3: Time complexity overview of main modules (default $B_f = 3$, $K = 128$, top-$k$=16).

| Module | Main Complexity | Description |
|---|---|---|
| rFFT | $\mathcal{O}(LC \log L)$ | One rFFT per channel |
| DynFBD | $\mathcal{O}(B_f KC)$ | Complex linear mapping, band projection |
| Channel Prior Mixer | $\mathcal{O}(Ck)$ | Aggregation after top-$k$ selection |
| Adaptive Fusion | $\mathcal{O}(n_{\text{heads}} K d^2)$ | Complex cross-attention on compressed tokens |
| Complex Encoder | $\mathcal{O}(n_{\text{layers}} d^2 K)$ | Two ComplexFullAttentionLayer layers |

Table 4: Ablation Study on the Interpretability Subset of Solar and Weather Datasets. We compare MSE performance and training Runtime (seconds per epoch). Note: The subset uses fewer samples (4,096) for rapid validation, resulting in different MSE scales compared to the full-dataset Main Results (Table 1).

| Config | Weather (21) | | Solar (137) | | Electricity (321) | |
|---|---|---|---|---|---|---|
| | MSE | Runtime (s) | MSE | Runtime (s) | MSE | Runtime (s) |
| FACT (concat) | **0.737** | 9.98 | **0.501** | 40.91 | **0.453** | 58.55 |
| **FACT (fusion)** | 0.783 | 10.51 | 0.523 | 17.17 | 0.468 | 10.23 |
| w/o DynFBD | 0.771 | **6.35** | 0.538 | **10.43** | 0.470 | **5.88** |
| w/o Channel Mix | 0.746 | 10.12 | 0.525 | 16.21 | 0.468 | 10.30 |
| $\lambda = 0.02$ | 0.744 | 10.49 | 0.522 | 16.99 | 0.468 | 10.24 |

## 5.4 INTERPRETABILITY VISUALIZATION

A key advantage of FACT is its transparency, which is intrinsic to the Interaction Module rather than dependent on a specific backbone. We visualize the patterns learned by the frontend modules on the Solar dataset in Figure 6.

The attention heatmaps (left), derived from the Adaptive Feature Fusion layer, reveal distinct frequency-band activations, indicating that the model selectively attends to specific periodic components. Since this attention mechanism is part of the feature alignment process, such fine-grained frequency interpretability is preserved even if the backend Encoder is replaced by an MLP.

The channel coherence map $\Gamma$ (center) captures the physical coupling between solar stations, aligning with geographical proximity. Guided gating trajectories (right) show how the model dynamically adjusts the importance of frequency bands during training, effectively filtering noise. These visualizations collectively demonstrate that FACT's explainability is rooted in its frequency-aware interaction design.

## 5.5 REGULARIZATION IMPACT

We investigate the impact of the regularization weight $\lambda$ (where $\lambda_{\text{coh}} = \lambda_{\text{phase}} = \lambda$) on the Weather dataset. As shown in Table 5, increasing the regularization strength from the default $\lambda = 0.01$ to $\lambda = 0.02$ leads to a significant improvement in MSE (from 0.783 to 0.744). This indicates that stronger enforcement of physical constraints (coherence and phase) can help the model generalize better by pruning spurious correlations.

Table 5: Sensitivity analysis of regularization weight $\lambda$ on Weather dataset (Interpretability Subset).

| $\lambda$ | MSE | Runtime (s) |
|---|---|---|
| 0.01 (Default) | 0.783 | 10.51 |
| 0.02 | **0.744** | **10.49** |

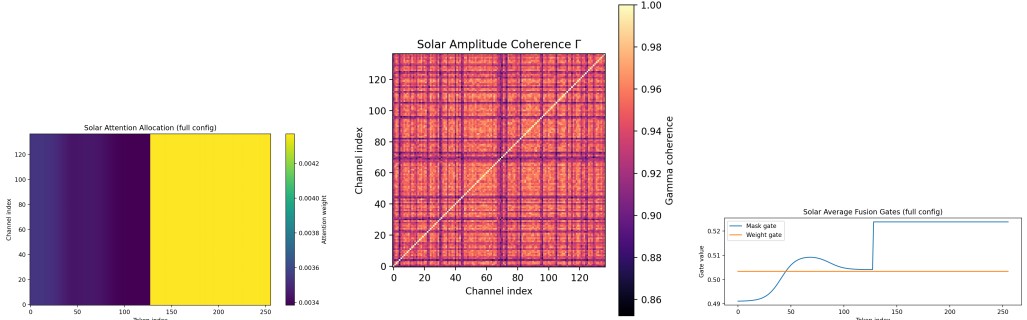

Figure 6: Interpretability on Solar: (Left) Attention heatmap showing frequency selection; (Center) Learned Amplitude Coherence Γ; (Right) Gating trajectories over training steps.

## 5.6 MODEL GENERALIZABILITY

Table 6: Model Generalizability: Performance and efficiency of FACT with different backbones ($L = 96, T = 96$). Lightweight backends (MLP/Linear) achieve comparable accuracy with significant speedups.

| Dataset | Backbone | MSE | MAE | Time (s/epoch) | Speedup |
|---|---|---|---|---|---|
| **Electricity** | Transformer | **0.145** | **0.243** | 99.37 | 1.0× |
| | MLP | 0.153 | 0.252 | 45.72 | 2.17× |
| | Linear | 0.155 | 0.254 | **43.14** | **2.30×** |
| **Solar** | Transformer | **0.192** | **0.236** | 74.59 | 1.0× |
| | MLP | 0.198 | 0.249 | 43.39 | 1.72× |
| | Linear | 0.211 | 0.264 | **39.84** | **1.87×** |

To verify the plug-in capability of our frequency frontend (Interaction Module), we evaluated three backends: Complex Transformer, Complex MLP, and Complex Linear. As shown in Table 6, replacing the heavy Transformer encoder with lightweight MLP or Linear layers results in only a marginal performance drop (e.g., $< 5\%$ MSE increase on Electricity) while delivering up to **2.3×** training speedup. On ETTh1, the FACT+MLP variant also achieved a competitive MSE of 0.456. This confirms that FACT's core benefits stem primarily from the frequency-aware interaction layer, which successfully disentangles signals for *any* backbone.

## 6 CONCLUSION

We propose FACT to resolve the tension between noise suppression and information preservation in multivariate time series forecasting by elevating interaction modeling from raw channels to fine-grained frequency components. By integrating Dynamic Frequency Band Decomposition with complex-valued, prior-guided interaction mechanisms, FACT effectively disentangles meaningful signals from noise while enforcing intrinsic interpretability through physical constraints. Extensive experiments validate FACT as a model-agnostic plug-in that yields consistent performance gains across diverse backbones (Transformer, MLP, Linear). While the current quadratic complexity poses scaling challenges for ultra-high-dimensional data, future integration with sparse attention or patching mechanisms promises to extend FACT's applicability, establishing a robust foundation for efficient, physically grounded forecasting systems. We believe this direction provides a new perspective for building efficient and interpretable time series systems in the future, and look forward to further validating its potential on larger-scale data and richer tasks.

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

## A  SYMBOL EXTENSIONS AND INFERENCE PSEUDOCODE

To facilitate reproduction, we supplement the key steps of FACT inference based on the symbols in the main text. The pseudocode mirrors the repository implementation, but we present it here using conceptual module names for clarity:

1. Input tensor $X \in \mathbb{R}^{B \times L \times C}$. If RevIN is enabled, execute $X \leftarrow \text{RevIN}(X)$ to obtain normalized representation; if reversible normalization is enabled, additionally cache mean and variance.

2. Compute $X_{\text{fft}} = \mathcal{F}_{\text{rfft}}(X)$, and pass it through the dynamic frequency-band preprocessor to obtain sparse frequency-domain tokens $\mathbf{Z}$, mask priors $\mathbf{M}$, and frequency-band weights $\boldsymbol{\omega}$.

3. Apply the frequency selector to smooth these weights, producing low-dimensional mask and weight summaries that will act as priors in later stages.

4. When channel mixing is enabled, estimate amplitude coherence $\gamma$ and phase priors $\phi$, construct mixing matrices and guided gating, and cache the resulting channel priors for regularization use.

5. Activate Adaptive Feature Fusion to re-weight frequency-domain representations through complex cross-attention informed by the aforementioned priors; otherwise, directly reuse the mixed spectrum $X_{\text{fft}}$.

6. Transform features back to the time domain and feed them into the chosen complex encoder (Transformer/MLP/Linear), obtaining prediction hidden states through the complex projection layer.

7. If reversible normalization or RevIN reverse process is enabled, restore original scale at output and extract the last $T$ step results.

## B  DATASET AND PREPROCESSING DETAILS

This paper follows the divisions published in SOFTS (Han et al., 2024), with related statistics in Table 8. Due to size limitations, the anonymous code package only includes Solar-137 examples. The loader implementation in the supplementary code package follows the considerations below:

- Data format: By default reads comma-separated floating-point text; for CSV files, skips the header row.

- Split strategy: Splits training/validation/test in chronological order according to 70/10/20, and fits the normalizer on the training set to prevent information leakage.

- Window parameters: the default window configuration $[96, 48, 96]$ is maintained as in the main experiments; the optional subsampling limit is set to 2000 rows for quick validation and can be disabled to load complete files.

- Temporal features: The anonymous release only supports the multivariate setting with standard time-encoding flags, consistent with Solar examples.

## C  TRAINING AND IMPLEMENTATION CONFIGURATION

Training uses the public entry point, with key hyperparameter default values as follows:

- Optimizer uses AdamW with learning rate $5 \times 10^{-4}$, combined with cosine annealing and linear warmup.

- Batch size 32, training epochs 10, early stopping patience 3. Interpretability subset scripts reduce the number of training epochs to three to shorten visualization generation time.

- Regularization coefficients $\lambda_{\text{coh}}$ and $\lambda_{\text{phase}}$ default to 0.01, and are skipped automatically when channel priors are unavailable.

- Complex attention defaults to two layers, hidden dimension 128, feedforward dimension 512; the token length produced by DynFBD is 128.

Table 7: FACT default hyperparameters (consistent with open-source implementation).

| Module | Key Parameters | Default Values / Notes |
|---|---|---|
| RevIN | use_revin, use_complex_revin, $\varepsilon$ | true, false, $1 \times 10^{-5}$ |
| Frequency Embedding | $d_{\text{model}}$, per-channel scale/bias | 128, learnable |
| BandPreprocessor | $B_f$, $K$, mask_proj_dim, weights_proj_dim | 3, 128, 16, 8 |
| Channel Prior Mixer | mixing_topk, $\tau$, mixing_strength, diag_bias, $\alpha$, $\beta$ | 16, 1.0, 0.1, 0.2, learnable |
| Guided Gating | gate_bias, gate_scale | 0.5, 0.5 |
| Adaptive Feature Fusion | $n_{\text{heads}}$, dropout, $\alpha$ | 8, 0.1, 0.7 |
| Complex Encoder | $e_{\text{layers}}$, $d_{\text{ff}}$ | 2 (main exp.) / 1 (interpretability subset), 512 |

Table 8: Dataset statistics (channels, horizons, splits, sampling rates).

| Dataset | Channels | Prediction Horizon $H$ | Data Split (Train, Val, Test) | Sampling Rate | Domain |
|---|---|---|---|---|---|
| ETTh1, ETTh2 | 7 | $\{96, 192, 336, 720\}$ | (8545, 2881, 2881) | Hourly | Electricity |
| ETTm1, ETTm2 | 7 | $\{96, 192, 336, 720\}$ | (34465, 11521, 11521) | 15min | Electricity |
| Weather | 21 | $\{96, 192, 336, 720\}$ | (36792, 5271, 10540) | 10min | Weather |
| ECL | 321 | $\{96, 192, 336, 720\}$ | (18317, 2633, 5261) | Hourly | Electricity |
| Traffic | 862 | $\{96, 192, 336, 720\}$ | (12185, 1757, 3509) | Hourly | Traffic |
| Solar-Energy | 137 | $\{96, 192, 336, 720\}$ | (36601, 5161, 10417) | 10min | Energy |
| PEMS03 | 358 | $\{12, 24, 48, 96\}$ | (15617, 5135, 5135) | 5min | Traffic |
| PEMS04 | 307 | $\{12, 24, 48, 96\}$ | (10172, 3375, 3375) | 5min | Traffic |
| PEMS07 | 883 | $\{12, 24, 48, 96\}$ | (16911, 5622, 5622) | 5min | Traffic |
| PEMS08 | 170 | $\{12, 24, 48, 96\}$ | (10690, 3548, 3548) | 5min | Traffic |

## D  ADDITIONAL EXPERIMENTAL RESULTS

Detailed interpretability metrics and regularization sensitivity statistics for Solar and Weather datasets are provided with accompanying CSV files, with values consistent with the main text analysis and can be directly accessed in the accompanying CSV tables.

## E  DATASET STATISTICS

Full statistics of the reused benchmarks are reported in Table 8.

## F  PRELIMINARIES (FULL)

### F.1  MULTIVARIATE LONG-TERM FORECASTING SETUP

Let the input sequence be $\mathbf{X} \in \mathbb{R}^{B \times L \times C}$. The target is to predict $\mathbf{Y} \in \mathbb{R}^{B \times T \times C}$ with loss $\mathcal{L}_{\text{forecast}} = \frac{1}{BCT} \sum_{b,t,c} (Y_{b,t,c} - \hat{Y}_{b,t,c})^2$.

### F.2  REAL FAST FOURIER TRANSFORM AND COMPLEX REPRESENTATION

Stack the time series as $\mathbf{X} \in \mathbb{R}^{L \times C}$, rFFT yields $\mathbf{X}_{\text{fft}} = \mathcal{F}_{\text{rfft}}(\mathbf{X}) \in \mathbb{C}^{F \times C}$ with $F = L/2 + 1$. For frequency $f$ and channel $c$, $\mathbf{X}_{\text{fft}}(f, c) = A(f, c)e^{i\theta(f,c)}$.

### F.3 DYNAMIC FREQUENCY-BAND DECOMPOSITION

For band $i$, the Gaussian weight is

$$\omega_i(f) = \frac{\exp\big(-(f-\mu_i)^2/(2\sigma_i^2)\big)}{\sum_{j=1}^{B_f} \exp\big(-(f-\mu_j)^2/(2\sigma_j^2)\big)}, \tag{13}$$

where $\mu_i, \sigma_i$ are learnable and $B_f = 3$ by default. Each band is compressed into $K$-dimensional tokens via complex linear projection.

### F.4 FREQUENCY SELECTION AND PROJECTION

Given $\mathbf{Z} \in \mathbb{C}^{K \times CB_f}$, the selector computes

$$\boldsymbol{\alpha} = \mathrm{softmax}\Big(\mathrm{Mean}_b(\sigma(|\mathbf{W}_1\mathbf{Z}|))\Big), \tag{14}$$

and projects it into mask/weight summaries $\mathbf{P}_{\mathrm{mask}} \in \mathbb{R}^{F \times d_m}$ and $\mathbf{P}_{\mathrm{weight}} \in \mathbb{R}^{K \times d_w}$ for subsequent priors and attention bias.

### F.5 CHANNEL CORRELATION AND PHASE PRIORS

Weighted amplitudes $\mathbf{A}_{c,f} = w_{\mathrm{eff}}(f)(A(f,c) - \mathrm{Mean}_f A(f,c))$ lead to

$$\gamma = \mathbf{A}\mathbf{D}^{-1}\mathbf{A}^\top, \tag{15}$$

where $\mathbf{D}$ normalizes $\gamma \in [-1,1]^{C \times C}$. Phase offsets summarize lead/lag:

$$\phi = \frac{\sin\boldsymbol{\theta}\,\cos\boldsymbol{\theta}^\top - \cos\boldsymbol{\theta}\,\sin\boldsymbol{\theta}^\top}{\max|\sin\boldsymbol{\theta}\,\cos\boldsymbol{\theta}^\top - \cos\boldsymbol{\theta}\,\sin\boldsymbol{\theta}^\top|}, \tag{16}$$

where $\sin\boldsymbol{\theta}, \cos\boldsymbol{\theta} \in \mathbb{R}^C$ are weighted by frequency.

### F.6 COMPLEX OPERATORS AND GUIDED GATING

For $\mathbf{z} = \mathbf{z}_r + \mathrm{i}\,\mathbf{z}_i$, a complex linear layer is

$$\mathrm{ComplexLinear}(\mathbf{z}) = (\mathbf{W}_r\mathbf{z}_r - \mathbf{W}_i\mathbf{z}_i) + \mathrm{i}(\mathbf{W}_i\mathbf{z}_r + \mathbf{W}_r\mathbf{z}_i). \tag{17}$$

Guided gating compresses weighted amplitudes to $[0,1]$ via

$$\mathbf{s} = \mathrm{Norm}_c(\mathrm{Mean}_f\, w_{\mathrm{eff}}(f)|\mathbf{X}_{\mathrm{fft}}(f,\cdot)|), \quad \mathbf{g} = \mathrm{gate\_bias} + \mathrm{gate\_scale} \cdot \mathrm{clip}(\mathbf{s}, 0, 1), \tag{18}$$

which stabilizes optimization and supports interpretability regularization.

## G ADDITIONAL VISUALIZATIONS

We provide additional interpretability visualizations for the Weather dataset in Figure 7, supplementing the Solar-137 analysis in the main text.

## H REPRODUCTION WORKFLOW SUMMARY

All figures and tables can be automatically generated through the auxiliary scripts shipped with the supplementary package. We keep the outline below at a high level and redact internal file names.

- Main results: run the standard FACT training recipe on Solar with DynFBD, channel mixing, and adaptive fusion enabled.
- Interpretability subset: execute the lightweight configuration on curated Solar/Weather subsets (4,096 samples, $e_{\mathrm{layers}} = 1$, 3 epochs).
- Attention heatmaps: post-process cached interpretability tensors to render attention and gating visualizations for Solar.

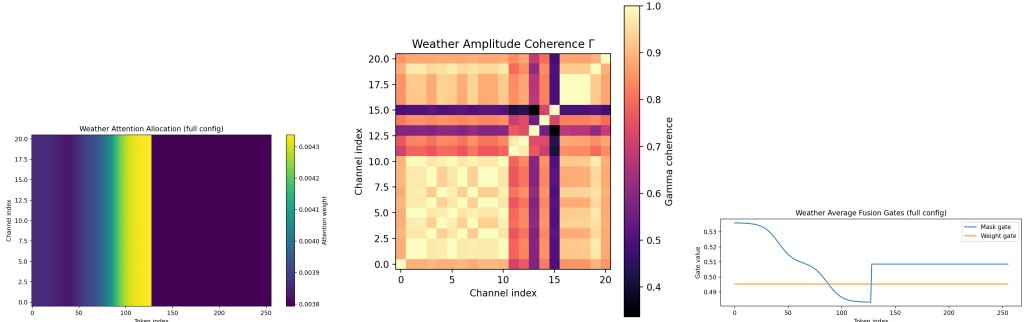

Figure 7: Attention, $\Gamma$ heatmaps and gating trajectories for Weather interpretability subset.

- Physical alignment: consolidate interpretability caches to compute $\Gamma/\Phi$ alignment statistics against meteorological variables.
- Regularization analysis: sweep coherence/phase regularization coefficients and export the summarized metrics.

The README in the supplementary scripts directory provides dataset-specific parameter examples that extend to domains such as Traffic and ECL.

## I    REPRODUCIBILITY CHECKLIST

High-level command reference for reproducing the main results and analyses:

- Main results: run the standard FACT training recipe with DynFBD, channel mixing, and adaptive fusion enabled.
- Interpretability subset: execute the lightweight configuration on Solar/Weather (4,096 samples, one encoder layer, three epochs).
- Heatmaps: post-process cached tensors to render attention and gating visualizations.
- Physical alignment: compute alignment between $\Gamma/\Phi$ and meteorological variables.
- Regularization: sweep $\lambda_{\mathrm{coh}}/\lambda_{\mathrm{phase}}$ and export summary tables.

## ETHICS STATEMENT

This research complies with the ICLR Code of Ethics. All experiments are based on public benchmarks.

The release and use of publicly available datasets respect their respective licenses and intended purposes. The proposed methodology is developed for scientific research and carries minimal risk of harmful applications. We acknowledge the broader concerns of fairness and bias in machine learning models, and we have taken steps to evaluate model robustness and to mitigate unintended discrimination.

No sensitive personal attributes were included in training or evaluation. This work does not involve conflicts of interest, unauthorized sponsorship, or activities that may compromise privacy, security, or research integrity.

## REPRODUCIBILITY STATEMENT

To facilitate the verification and extension of our work, we provide the following resources:

- **Code Availability:** The complete implementation is available at: `https://anonymous.4open.science/r/FACT`

- **Datasets:** All experiments are based on public benchmarks (ETT, Traffic, Electricity, Weather, Solar-Energy).
- **Key Components:** The core innovations include:
  - Dynamic Frequency-Band Decomposition (DynFBD)
  - ChannelPriorMixer for amplitude-phase priors
  - Complex cross-attention fusion
- **Training Setup:** We employ standard hyperparameters (learning rate=5e-4, batch size=32) alongside coherence and phase regularization.

We confirm that all reported results can be reproduced with minimal error using the provided resources and configuration.

## LLM USAGE

Large Language Models (LLMs) were used exclusively for polishing the language and writing of this manuscript. The LLM contributed neither to the research conception nor to the core intellectual content. We bear full responsibility for the work presented herein.

