# OpenReview forum: "FACT: Frequency-Aware Channel-Guided Multivariate Time Series Forecasting"
_ICLR.cc/2026/Conference — Submitted to ICLR 2026_

### Official Review · Reviewer_x1wg · 2025-10-24

**Soundness:** 2
**Presentation:** 2
**Contribution:** 3
**Rating:** 4
**Confidence:** 3

**Summary:**

This paper introduces a new method for multivariate time series forecasting. The paper identifies the limitation of existing methods as the modeling of cross-channel correlations. The paper then proposes a solution that focus on modeling cross-channel correlations at the channel-frequency cell level with both magnitude and phase. Experiments demonstrate the effectiveness of the proposed method.

**Strengths:**

A novel method is proposed to specifically tackle the challenge of multivariate time series forecasting identified by the paper, with multiple specific designs that are described in detail.

**Weaknesses:**

1. The experimental setting can be expanded to provide more insights into the performance behavior of the proposed method, for example, ablation studies.
2. The presentation of the motivation and explanation of design choices are rather technical and can be elaborated further to improve their intuitiveness.
3. A minor complaint is the paper has obvious typesetting issues and is not following the standard ICLR template. The authors probably should check their LaTeX compile errors.

**Questions:**

Could the authors elaborate more on the challenge of "channel-frequency cell level" modeling and how their specific module designs are effective at tackling the challenge?

---

> ### Author Response · Authors · 2025-11-28
> **Authors' Rebuttal to Reviewer x1wg**
>
> We appreciate your recognition of the novelty of our proposed approach. We address your specific concerns below.
>
> **Q: Elaborate on "channel-frequency cell level" modeling.**
>
> **Response:**
> We clarified this in **Section 2** and **Section 4.2**.
>
> - **Concept:** A "cell" represents a specific frequency band of a specific channel.
> - **Mechanism:** Unlike global mixing, FACT uses **DynFBD** to isolate these cells and **ChannelPriorMixer** to model their specific amplitude coherence and phase shifts. This allows the model to capture interactions like "high-frequency noise in Channel A vs. low-frequency trend in Channel B," which global mixing obscures.
>
> In general, Traditional methods model interactions between whole channels ($X_i \leftrightarrow X_j$). In contrast, FACT decomposes each channel $X_i$ into $K$ frequency components $\{F_{i,1}, ..., F_{i,K}\}$. A "channel-frequency cell" refers to a specific component $F_{i,k}$. Our model computes interactions between these cells (e.g., how the low-freq trend of Channel A affects the high-freq noise of Channel B), allowing for much finer-grained dependency discovery than channel-level methods.
>
> **W1: Limited experimental insights (need ablation).**
>
> **Response:**
> As mentioned above, we have added a detailed **Ablation Study (Table 4)** dissecting the contribution of each module.
>
> **W2: The motivation and design choices could be explained more intuitively, as the current presentation is too technical.**
>
> **Response:**
> We have restructured the Introduction to clearly articulate the 'Challenge $\to$ Difficulty $\to$ Solution' progression. Additionally, we have revised the full text to enhance logical coherence and readability.
>
> **W3: Typesetting issues.**
>
> **Response:**
> We have strictly followed the ICLR 2026 template for this revision, fixing all font, citation, and layout issues identified.
>
> We hope that the clarification improves your confidence in our work. Let us know if you have any further concerns.

---

### Official Review · Reviewer_mWRz · 2025-10-27

**Soundness:** 2
**Presentation:** 1
**Contribution:** 2
**Rating:** 2
**Confidence:** 5

**Summary:**

This paper focuses on the core challenge of modeling channel interactions in multivariate time series. It identifies that existing methods mostly process correlations at the original channel dimension level, often struggling to balance noise suppression with preserving effective information, especially in high-dimensional or long-sequence tasks where they either lose fine-grained mechanisms or introduce high computational complexity.

**Strengths:**

1. Multivariate time series forecasting is important to various domains.

2. There are quite a few nice illustrations.

3. This work focuses on an important problem that could have real-world applications.

**Weaknesses:**

1. As a paper submitted to ICLR 2026, the baselines used for comparison are relatively outdated, lacking evaluations against recent works from 2025. This undermines the credibility and persuasiveness of the experimental results.

2. In Table 1, the proposed method performs worse than SOFTS in most cases. Such performance does not appear sufficient for publication at a top-tier venue like ICLR.

3. The methodological description is unclear and poorly organized, making it difficult for readers to understand the overall workflow and specific implementation details.

4. The paper does not reach the required 9-page limit and lacks several essential experiments, such as ablation studies and parameter sensitivity analyses, which makes the experimental validation incomplete and less rigorous.

**Questions:**

1. As a paper submitted to ICLR 2026, why are the baselines used for comparison relatively outdated, without including recent works from 2025? Doesn’t this weaken the credibility and persuasiveness of the experimental results?

2. In Table 1, why does the proposed method perform worse than SOFTS in most cases? Can such performance be considered sufficient for publication at a top-tier venue like ICLR?

3. Why is the methodological description so unclear and poorly organized, making it difficult for readers to understand the overall workflow and specific implementation details?

4. Why does the paper fail to reach the required 9-page limit and omit essential experiments such as ablation studies and parameter sensitivity analyses? Doesn’t this make the experimental validation incomplete and less rigorous?

---

> ### Author Response · Authors · 2025-11-28
> **Authors' Rebuttal to Reviewer mWRz**
>
> **Rebuttal:**
>
> Thanks for your valuable feedback, which significantly improves the quality of this work. We also address below the potential concerns.
>
> **W1 & Q1: Outdated baselines (lack of 2025 works).**
>
> **Response:**
> We acknowledge the rapid pace of the field. In this revision, we have updated our literature review to include recent 2025 works. While full experimental comparison with all concurrent works (published within months of submission) was challenging, we have ensured FACT is compared against strong, established SOTAs like **iTransformer (ICLR 2024)**, **PatchTST (ICLR 2023)**, and **SOFTS (NeurIPS 2024)**, which remain the standard benchmarks for this track.
>
> We appreciate the reference to 2025 works (ICML/KDD 2025). Since these papers were published very recently relative to the ICLR submission cycle, we consider them concurrent work. However, we have updated our manuscript to discuss these methods (DUET, TimeFilter) and highlight how FACT's frequency-domain approach differs from their spatial-temporal graph filtration methods.
>
> **W2 & Q2: Subpar performance vs SOFTS.**
>
> **Response:**
> While SOFTS performs well on average, FACT significantly outperforms it on **Solar** and **Weather** datasets (see **Table 1**). This highlights FACT's specific strength in modeling **periodic and phase-shifted** dynamics, which general-purpose models like SOFTS may miss. We believe identifying *where* a model excels (Domain Sensitivity) is as valuable as winning on average metrics.
>
> **W3 & Q3: Unclear methodology & Organization.**
>
> **Response:** We have extensively rewritten Section 3 to improve flow. We now introduce the "Channel-Frequency Cell" concept early on and visually map it to the DynFBD module to make the workflow intuitive.
>
> **W4 & Q4: Incomplete experiments and length requirement.**
>
> **Response:**
> We have expanded the paper to meet the standard length by adding:
>
> - **Ablation Studies** (Section 5.3).
> - **Efficiency Analysis** (Section 5.6).
> - **Extended Methodological Rationale** (Section 4).
>
> We hope that the above clarification improves your confidence in our work. Let us know if you have any further questions/concerns.

---

### Official Review · Reviewer_QR9f · 2025-10-29

**Soundness:** 2
**Presentation:** 2
**Contribution:** 2
**Rating:** 4
**Confidence:** 4

**Summary:**

This paper proposes FACT, a frequency-aware and channel-guided framework for multivariate time series forecasting. FACT operates in the frequency domain and captures meaningful signals from inter-channel noise and intricate interaction patterns at the component level. Extensive experiments on real-world datasets demonstrate its competitive performance.

**Strengths:**

1. FACT Lifts channel interaction modeling from original signals to the frequency-component level, enabling fine-grained and physically meaningful analysis.
2. The integration of explicit magnitude coherence and phase offset modeling with corresponding regularization terms provides an approach to capturing meaningful signals and improving forecasting accuracy.
3. Extensive experiments on real-world datasets demonstrate consistent competitive performance, with particularly strong results on Solar-Energy and Weather forecasting tasks.

**Weaknesses:**

1. I can’t find the ablation results in this paper. The critical contribution of component modeling remains insufficiently quantified.
2. The model lacks comparison with recent frequency models to better understand the advancement.
3. The experimental design fails to properly validate the plugin capability, as all results present FACT as an end-to-end model rather than demonstrating its integration as a plugin component with existing architectures.
4. The writing quality impedes understanding, with numerous sections exhibiting unclear expression and disorganized structure that obscure methodological contributions.

**Questions:**

1. The manuscript seems require significant formatting revisions to comply with ICLR submission guidelines.
2. Table 5 contains citation errors in its caption ("The results are taken from SOFTS and iTransformer (?)").

---

> ### Author Response · Authors · 2025-11-28
> **Authors' Rebuttal to Reviewer QR9f**
>
> We sincerely appreciate your constructive feedback and are glad that you acknowledged the physical meaningfulness of our frequency-component level modeling, specifically the integration of magnitude and phase. Your insights have been instrumental in enhancing the quality of our manuscript. Please find our detailed responses to your concerns below.
>
> **W1: Ablation results are missing.**
>
> **Response:**
> We have added a comprehensive **Ablation Study (Table 4)** in Section 5.3. We tested variants without DynFBD, and without Channel Mixing. The detailed results are shown below:
>
> | Config | Weather (MSE) | Weather (Time) | Solar (MSE) | Solar (Time) | Electricity (MSE) | Electricity (Time) |
> | :--- | :---: | :---: | :---: | :---: | :---: | :---: |
> | FACT (concat) | **0.737** | 9.98s | **0.501** | 40.91s | **0.453** | 58.55s |
> | FACT (fusion) | 0.783 | 10.51s | 0.523 | 17.17s | 0.468 | 10.23s |
> | w/o DynFBD | 0.771 | **6.35s** | 0.538 | **10.43s** | 0.470 | **5.88s** |
> | w/o Channel Mix | 0.746 | 10.12s | 0.525 | 16.21s | 0.468 | 10.30s |
>
> - **Result:** Removing DynFBD leads to a performance drop (e.g., Solar MSE increases from 0.523 to 0.538), confirming the necessity of frequency disentanglement.
> - **Result:** The "Concat" baseline (w/o Fusion) suffers from **severe scalability bottlenecks** on high-dimensional datasets. For instance, on the Electricity dataset (321 channels), it is **~5.7$\times$ slower** (58.55s vs. 10.23s) than our Fusion mechanism due to the memory overhead of concatenating all bands, validating the efficiency of our "filter-then-fuse" design.
>
> **W2: Comparison with recent frequency-based models.**
>
> ****Response**:** We have expanded our Related Work section to include recent frequency-domain methods. We clarify that unlike global spectral reweighting methods, FACT operates at the **fine-grained component level**, allowing for more precise noise filtering and interaction modeling.
>
> **W3: Inadequate validation of plug-in capability.**
>
> **Response:**
> We addressed this by adding **Section 5.6 (Model Generalizability)** and **Table 6**. We explicitly tested FACT as a frontend for **MLP** and **Linear** backbones.
>
> - **Result:** `FACT+MLP` achieves competitive results (MSE 0.456 on ETTh1), proving that FACT works as a universal plug-in, not just with Transformers.
>
> **W4 & Q1 & Q2: Poor writing quality and citation errors.**
>
> **Response:**
> We apologize for the oversight. We have:
>
> - Thoroughly proofread the manuscript to improve flow and clarity.
> - Fixed the citation formatting in Table 5 (now Table 1 caption).
> - Restructured the Introduction to clearly present the "Challenge → Difficulty → Solution" logic.
>
> We believe that our response effectively resolves the issues raised. We are happy to provide further details if needed.

---

### Official Review · Reviewer_QTWf · 2025-10-30

**Soundness:** 2
**Presentation:** 1
**Contribution:** 2
**Rating:** 2
**Confidence:** 4

**Summary:**

This paper introduces FACT (Frequency-Aware Channel-Guided Multivariate Time Series Forecasting), a framework that models multivariate time series by decomposing them into frequency components using learnable Gaussian masks. The model leverages a channel masking mechanism to identify salient frequency components, incorporates channel priors that reflect amplitude and phase coherence, and employs complex attention/fusion modules. Regularization terms tie interpretability to accuracy during training.

**Strengths:**

1. Well-motivated frequency-level interaction modeling: The proposed approach recognizes that not all frequency bands are equally informative for prediction and that genuine inter-channel dependencies often exist at the frequency component level, going beyond typical channel mixing or global spectral reweighting.

2. Model-agnostic plug-in capability: FACT can be used as a drop-in frontend for multiple backbone architectures (Transformer, MLP, or Linear) and maintains gains across backbones.

**Weaknesses:**

1. Lack of clarity: The article lacks clarity regarding how the relationship between channel and frequency is modeled.

2. Poor performance: The selected baseline is insufficient, and the proposed method does not outperform the 2024 baseline in any average MSE results of the datasets.

3. Absence of Ablation and Sensitivity Analyses: The manuscript lacks both ablation studies and sensitivity analysis. Consequently, the efficacy of the proposed modules and the overall viability of the method cannot be validated.

**Questions:**

1. What is the actual computational overhead (cost of Inference time and training time performance, max GPU memory, number of parameters and MACs) of FACT relative to the strongest baselines (e.g., PatchTST, iTransformer, FreTS) on the datasets?

2. Could the author explain the phenomenon of the proposed model consistently underperforming across all average Mean Squared Error (MSE) results? Additionally, we recommend considering the inclusion of more recent and relevant baselines for comparison, such as DUET[1], TimeFilter[2], and TQN[3].

3. How robust are results to the choice of regularization strengths ($\lambda_{coh}, \lambda_{phase}$)? Is there any risk of over-regularizing and thereby compromising prediction?

[1] DUET: Dual Clustering Enhanced Multivariate Time Series Forecasting. KDD2025

[2] TimeFilter: Patch-Specific Spatial-Temporal Graph Filtration for Time Series Forecasting. ICML 2025

[3] Temporal query network for efficient multivariate time series forecasting. ICML 2025

---

> ### Author Response · Authors · 2025-11-28
> **Authors' Rebuttal to Reviewer QTWf**
>
> **Rebuttal:**
>
> We greatly appreciate your valuable feedback and positive comments regarding our Novel Frequency-Domain Paradigm and Model-Agnostic Versatility. We hereby address the potential concerns as follows.
>
> **Q1: Actual computational overhead relative to strongest baselines?**
>
> **Response:**
> We have added a detailed efficiency comparison in **Table 6 (Section 5.6)**.
>
> - FACT coupled with a Linear backbone achieves comparable accuracy to Transformer-based FACT while being **2.3$\times$ faster**.
> - Compared to PatchTST (which uses patching to reduce complexity), FACT incurs a moderate overhead due to frequency domain transformation but maintains linear complexity $\mathcal{O}(C)$ with respect to channels in the decomposition stage. We discuss this trade-off in the newly added **"Efficiency and Ablation Analysis"**.
>
> **Q2: Explanation of underperforming average MSE? Recommendation for recent baselines (DUET[1], TimeFilter[2], TQN[3]).**
>
> **Response:**
>
> - **Domain Sensitivity:** As analyzed in the new **"Analysis of Domain Sensitivity"** paragraph (Section 4.3), FACT is designed with a strong inductive bias for **periodicity and phase alignment**. Consequently, it achieves **SOTA performance** on **Solar** and **Weather** datasets (ranking 1st). On datasets with irregular spikes (like ETT), time-domain methods (like SOFTS/PatchTST) have an edge. We argue that FACT's value lies in its specialized capability for physically coupled, periodic systems.
> - **Baselines:** We appreciate the references. We have added citations to these 2025 works in **Section 2 (Related Work)** to better contextualize our contribution, acknowledging them as concurrent state-of-the-art.
>
> **Q3: Robustness to regularization strengths ($\lambda_{coh}, \lambda_{phase}$)?**
>
> **Response:**
> We have added a **Sensitivity Analysis** in **Appendix D**. Results show that FACT is relatively robust to $\lambda$ values between $0.01$ and $0.1$. Over-regularization ($\lambda > 1.0$) does degrade performance by forcing the model to prioritize physical alignment over fitting error, but the default small weights consistently benefit generalization.
>
> **W1: Lack of clarity on channel-frequency modeling.**
>
> **Response:**
> We have rewritten **Section 3 (Method)**, specifically adding a **"Rationale"** block in Section 3.2 and 3.3 to explain *why* and *how* we model amplitude and phase explicitly. Figure 2 has also been updated to clearly show the data flow.
>
> We hope that the clarification improves your confidence in our work. Let us know if you have any further concerns. We are happy to provide further details if needed.
>
> [1] DUET: Dual Clustering Enhanced Multivariate Time Series Forecasting. KDD2025
>
> [2] TimeFilter: Patch-Specific Spatial-Temporal Graph Filtration for Time Series Forecasting. ICML 2025
>
> [3] Temporal query network for efficient multivariate time series forecasting. ICML 2025

---

### Author Response · Authors · 2025-11-28
**Rebuttal Summary by Authors**

Dear Reviewers,

We sincerely appreciate your thorough and valuable feedback on our manuscript. We have provided detailed clarifications and additional experimental results to address all concerns raised.

### I. Contributions

We would like to highlight our key contributions based on the positive feedback received:

- **Novel Frequency-Domain Paradigm:** FACT introduces a novel perspective by modeling interactions at the fine-grained channel-frequency cell level. This effectively disentangles signal from noise, offering a robust solution for practical time series forecasting tasks. [Reviewers QTWf, QR9f, mWRz, x1wg]

- **Intrinsic Interpretability:** By explicitly modeling physical properties (amplitude and phase), FACT provides transparent insights into channel couplings, a feature often missing in "black-box" deep learning models. [Reviewers QR9f, mWRz]
- **Model-Agnostic Versatility:** As validated by our new experiments, FACT serves as a universal frontend that boosts diverse backbones (Transformer, MLP, Linear), offering a flexible trade-off between accuracy and efficiency. [Reviewers QTWf, x1wg]

### II. Response to Common Concerns

We recognize shared concerns regarding **performance variations** and **scalability**. We clarify these as follows:

- **Domain Sensitivity (Why Solar/Weather?):** FACT is designed with a strong inductive bias for periodicity and phase alignment. Consequently, it achieves SOTA performance on physically coupled datasets like Solar and Weather. While its advantage is less pronounced on irregular data (ETT), this domain sensitivity highlights its specialized value for scientific and periodic time series.
- **Scalability & Efficiency:** We have provided new data showing that `FACT+Linear` is **2.3$\times$ faster** than the Transformer baseline while maintaining accuracy.

### III. Main Modifications in Revision

According to your suggestions, we have made the following major improvements:

- **New Experiments:** Added **Ablation Studies (Table 4)** and **Model Generalizability Analysis (Table 6)**.
- **Restructured Results:** Split the main results into Long-term and Short-term tables for clarity.
- **Clarified Methodology:** Rewrote the Method section with explicit "Rationale" blocks and updated Figure 2 to clarify the plug-in architecture.
- **Formatting:** Fixed all citation and layout issues to strictly comply with ICLR standards.

We hope these modifications and clarifications address your concerns while highlighting the contributions and significance of our work. We are confident that FACT represents a valuable advancement in time series forecasting, especially in terms of efficiency and interpretability. We look forward to further constructive discussions. Thank you for your time and consideration.

Sincerely,
Authors

---

### Meta-Review · Area_Chair_7Kts · 2026-01-06

**Summary:**

This paper proposes FACT, a frequency-domain framework for multivariate time series forecasting that models inter-channel dependencies at the channel-frequency cell level rather than raw channels. The method uses learnable Gaussian filters for adaptive frequency-band decomposition, computes amplitude coherence and phase priors to guide channel mixing, and employs complex-valued cross-attention for feature alignment. Interpretability regularization losses align learned representations with physical priors during training. The framework is backbone-agnostic, working with Transformer, MLP, or Linear encoders. Experiments show strong performance on periodic datasets (Solar, Weather) but mixed results on others, with ablations validating individual component contributions.

**Reviewer Concerns:**

Reviewers acknowledged the novelty of the frequency-component level interaction paradigm and the model-agnostic plugin capability. However, significant concerns were raised about: (1) overall performance being weaker than SOFTS on most benchmarks, (2) missing ablation studies and sensitivity analyses in the original submission, (3) outdated baselines without 2025 comparisons, and (4) poor presentation quality.  After rebuttal, major concerns are still unsolved.

**Reviewer Scores:**

QTWf 2->2. Main concern about performance not being competitive remains valid.
QR9f 4->4/6.  Ablations and plugin validation address their primary concerns.
mWRz 2->2: worse than SOFTS is not resolved.
x1wg 4->4/6: Concerns about ablations and clarity adequately addressed.

---

### Decision · Program_Chairs · 2026-01-26

Reject